# Self-Supervised Video Representation Learning with Constrained Spatiotemporal Jigsaw

## Abstract

This paper proposes a novel pretext task for self-supervised video representation learning by exploiting spatiotemporal continuity in videos. It is motivated by the fact that videos are spatiotemporal by nature and a representation learned to detect spatiotemporal continuity/discontinuity is thus beneficial for downstream video content analysis tasks. A natural choice of such a pretext task is to construct spatiotemporal (3D) jigsaw puzzles and learn to solve them. However, this task turns out to be intractable. We thus propose Constrained Spatiotemporal Jigsaw (CSJ) whereby the 3D jigsaws are formed in a constrained manner to ensure that large continuous spatiotemporal cuboids exist in a shuffled clip to provide sufficient cues for the model to reason about the continuity. With the constrained jigsaw puzzles, instead of solving them directly, which could still be extremely hard, we carefully design four surrogate tasks that are more solvable but meanwhile still ensure that the learned representation is sensitive to spatiotemporal continuity at both the local and global levels. Extensive experiments show that our CSJ achieves state-of-the-art on two downstream tasks across various benchmarks.

## 1 Introduction

Self-supervised learning (SSL) has achieved tremendous successes recently for static images (He et al., 2020; Chen et al., 2020) and shown to be able to outperform supervised learning on a wide range of downstream image understanding tasks. However, such successes have not yet been reproduced for videos. Since different SSL models differ mostly on the pretext tasks employed on the unlabeled training data, designing pretext tasks more suitable for videos is the current focus for self-supervised video representation learning (Han et al., 2020; Wang et al., 2020).

Videos are spatiotemporal data and spatiotemporal analysis is the key to many video content understanding tasks. A good video representation learned from the self-supervised pretext task should therefore capture discriminative information jointly along both spatial and temporal dimensions. It is thus somewhat counter-intuitive to note that most existing SSL pretext tasks for videos do not explicitly require joint spatiotemporal video understanding. For example, some spatial pretext tasks have been borrowed from images without any modification (Jing et al., 2018), ignoring the temporal dimension. On the other hand, many recent video-specific pretext tasks typically involve speed or temporal order prediction (Lee et al., 2017; Wei et al., 2018; Benaim et al., 2020; Wang et al., 2020), *i.e.*, operating predominately along the temporal axis.

A natural choice for a spatiotemporal pretext task is to solve 3D jigsaw puzzles, whose 2D counterpart has been successfully used for images (Noroozi & Favaro, 2016). Indeed, solving 3D puzzles requires the learned model to understand spatiotemporal continuity, a key step towards video content understanding. However, directly solving a 3D puzzle turns out to be intractable: a puzzle of $3\times3\times3$ pieces (the same size as a Rubik's cube) can have 27! possible permutations. Video volume even in a short clip is much larger than that. Nevertheless, the latest neural sorting models (Paumard et al., 2020; Du et al., 2020) can only handle permutations a few orders of magnitude less, so offer no solution. This is hardly surprising because such a task is daunting even for humans: Most people would struggle with a standard Rubik's cube, let alone a much larger one.

In this paper, we propose a novel Constrained Spatiotemporal Jigsaw (CSJ) pretext task for self-supervised video representation learning. The key idea is to form 3D jigsaw puzzles in a constrained manner so that it becomes solvable. This is achieved by factorizing the permutations (shuffling)

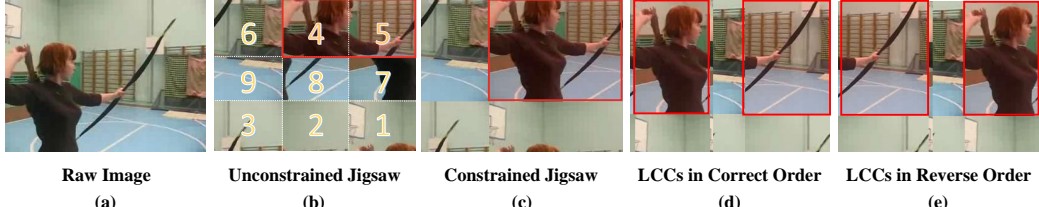

| Raw Image | Unconstrained Jigsaw | Constrained Jigsaw | LCCs in Correct Order | LCCs in Reverse Order |
|:---:|:---:|:---:|:---:|:---:|
| (a) | (b) | (c) | (d) | (e) |

Figure 1: Illustration of our constrained jigsaw and the surrogate pretext tasks using an image example (only spatial for clarity). Our constrained jigsaw can be easily extended to the spatiotemporal domain as done in this work. (a): The raw image. (b),(c): Comparing an unconstrained puzzle (b) and our constrained one (c), it is clear that ours is much more continuous (hence interpretable) reflected by the size of the largest continuous cuboids (LCCs, rectangles in images here) shown in red. (d),(e): Illustration of the importance of the relative order of the top-2 LCCs for determining the global continuity level of the shuffled image. (d) and (e) have the same top-2 LCCs, but only (d) keeps the correct relative order between them. Locating these LCCs and predicting their relative order are thus the key objectives of our surrogate tasks.

into the three spatiotemporal dimensions and then applying them sequentially. This ensures that for a given video clip, large continuous spatiotemporal cuboids exist after the constrained shuffling to provide sufficient cues for the model to reason about spatiotemporal continuity (see Fig. 1(b)(c)). Such large continuous cuboids are also vital for human understanding of video as revealed in neuroscience and visual studies (Stringer et al., 2006; Chen et al., 2019). Even with the constrained puzzles, solving them directly could still be extremely hard. Consequently, instead of directly solving the puzzles (*i.e.*, recovering the permutation matrix so that each piece can be put back), four surrogate tasks are carefully designed. They are more solvable but meanwhile still ensure that the learned representation is sensitive to spatiotemporal continuity at both the local and global levels. Concretely, given a video clip shuffled with our constrained permutations, we make sure that the top-2 largest continuous cuboids (LCCs) dominate the clip volume. The level of continuity in the shuffle clip as a whole is thus determined mainly by the volumes of these LCCs, and whether they are at the right order (see Fig. 1(d)(e)) both spatially and temporally. Our surrogate tasks are thus designed to locate these LCCs and predict their order so that the model learned with these tasks can be sensitive to spatiotemporal continuity both locally and globally.

Our main contributions are three-fold: (1) We introduce a new pretext task for self-supervised video representation learning called Constrained Spatiotemporal Jigsaw (CSJ). To our best knowledge, this is the first work on self-supervised video representation learning that leverages spatiotemporal jigsaw understanding. (2) We propose a novel constrained shuffling method to construct easy 3D jigsaws containing large LCCs. Four surrogate tasks are then formulated in place of the original jigsaw solving tasks. They are much more solvable yet remain effective in learning spatiotemporal discriminative representations. (3) Extensive experiments show that our approach achieves state-of-the-art on two downstream tasks across various benchmarks.

## 2 RELATED WORK

**Self-supervised Learning with Pretext Tasks** Self-supervised learning (SSL) typically employs a pretext task to generate pseudo-labels for unlabeled data via some forms of data transformation. According to the transformations used by the pretext task, existing SSL methods for video presentation learning can be divided into three categories: **(1) Spatial-Only Transformations**: Derived from the original image domain (Gidaris et al., 2018), Jing et al. (2018) leveraged the spatial-only transformations for self-supervised video presentation learning. **(2) Temporal-Only Transformations**: Misra et al. (2016); Fernando et al. (2017); Lee et al. (2017); Wei et al. (2018) obtained shuffled video frames with the temporal-only transformations and then distinguished whether the shuffled frames are in chronological order. Xu et al. (2019) chose to shuffle video clips instead of frames. Benaim et al. (2020); Yao et al. (2020); Jenni et al. (2020) exploited the speed transformation via determining whether one video clip is accelerated. **(3) Spatiotemporal Transformations**: There are only a few recent approaches (Ahsan et al., 2019; Kim et al., 2019) that leveraged both spatial and temporal transformations by permuting 3D spatiotemporal cuboids. However, due to the aforementioned

intractability of solving the spatiotemporal jigsaw puzzles, they only leveraged either temporal or spatial permutations as training signals, *i.e.*, they exploited the two domains independently. Therefore, *no true spatiotemporal permutations* have been considered in Ahsan et al. (2019); Kim et al. (2019). In contrast, given that both spatial appearances and temporal relations are important cues for video representation learning, the focus of this work is on investigating how to exploit the spatial and temporal continuity jointly for self-supervised video presentation learning. To that end, our Constrained Spatiotemporal Jigsaw (CSJ) presents the first spatiotemporal continuity based pretext task for video SSL, thanks to a novel constrained 3D jigsaw and four surrogate tasks to reason about the continuity in the 3D jigsaw puzzles without solving them directly.

**Self-supervised Learning with Contrastive Learning**    Contrastive learning is another self-supervised learning approach that has become increasingly popular in the image domain (Misra & Maaten, 2020; He et al., 2020; Chen et al., 2020). Recently, it has been incorporated into video SSL as well. Contrastive learning and transformation based pretext tasks are orthogonal to each other and often combined in that different transformed versions of a data sample form the positive set used in contrastive learning. In El-Nouby et al. (2019); Knights et al. (2020); Qian et al. (2020); Wang et al. (2020); Yang et al. (2020), the positive/negative samples were generated based on temporal transformations only. In contrast, some recent works (Han et al., 2019; 2020; Zhuang et al., 2020) leveraged features from the future frame embeddings or with the memory bank (Wu et al., 2018). They modeled spatiotemporal representations using only contrastive learning without transformations. Contrastive learning is also exploited in one of our surrogate pretext tasks. Different from existing works, we explore the spatiotemporal transformations in the form of CSJ and employ contrastive learning to distinguish different levels of spatiotemporal continuity in shuffled jigsaws. This enables us to learn more discriminative spatiotemporal representations.

## 3    CONSTRAINED SPATIOTEMPORAL JIGSAW

### 3.1    PROBLEM DEFINITION

The main goal of self-supervised video representation learning is to learn a video feature representation function $f(\cdot)$ without using any human annotations. A general approach to achieving this goal is to generate a supervisory signal $y$ from an unlabeled video clip $x$ and construct a pretext task $P$ to predict $y$ from $f(x)$. The process of solving the pretext task $P$ encourages $f(\cdot)$ to learn discriminative spatiotemporal representations.

The pretext task $P$ is constructed typically by applying to a video clip a transformation function $t(\cdot; \boldsymbol{\theta})$ parameterized by $\boldsymbol{\theta}$ and then automatically deriving $y$ from $\boldsymbol{\theta}$, *e.g.*, $y$ can be the type of the transformation. Based on this premise, $P$ is defined as the prediction of $y$ using the feature map of the transformed video clip $f(\widetilde{x})$, *i.e.*, $P : f(\widetilde{x}) \to y$, where $\widetilde{x} = t(x; \boldsymbol{\theta})$. For example, in Lee et al. (2017), $t(\cdot; \boldsymbol{\theta})$ denotes a temporal transformation that permutes the four frames of video clip $x$ in a temporal order $\boldsymbol{\theta}$, $\widetilde{x} = t(x; \boldsymbol{\theta})$ is the shuffled clip, and the pseudo-label $y$ is defined as the permutation order $\boldsymbol{\theta}$ (*e.g.*, 1324, 4312, etc.). The pretext task $P$ is then a classification problem of 24 categories because there are $4! = 24$ possible orders.

### 3.2    CONSTRAINED PERMUTATIONS

Solving spatiotemporal video jigsaw puzzles seems to be an ideal pretext task for learning discriminative representation as it requires an understanding of spatiotemporal continuity. After shuffling the pixels in a video clip using a 3D permutation matrix, the pretext task is to recover the permutation matrix. However, as explained earlier, this task is intractable given even moderate video clip sizes. Our solution is to introduce constraints on the permutations. As a result, a new pretext task $P_{\text{CSJ}}$ based on **Constrained Spatiotemporal Jigsaw** (see Fig. 2(a)) is formulated, which is much easier to solve than a random/unconstrained jigsaw.

Specifically, our goal is to introduce constraints to the permutations so that the resultant shuffled video clip is guaranteed to have large continuous cuboids (see Fig. 2(a)). Similar to humans (Stringer et al., 2006), having large continuous cuboids is key for a model to understand a 3D jigsaw and therefore to have any chance to solve it. Formally, the volume of a shuffled video clip $\widetilde{x}$ are denoted as $\{T, H, W\}$, measuring its sizes along the temporal, height, and width dimensions, respectively. A cuboid is defined as a crop of $\widetilde{x}$: $\boldsymbol{c} = \widetilde{x}_{t_1:t_2, h_1:h_2, w_1:w_2}$, where $t_1, t_2 \in \{1, 2, \ldots, T\}, h_1, h_2 \in$

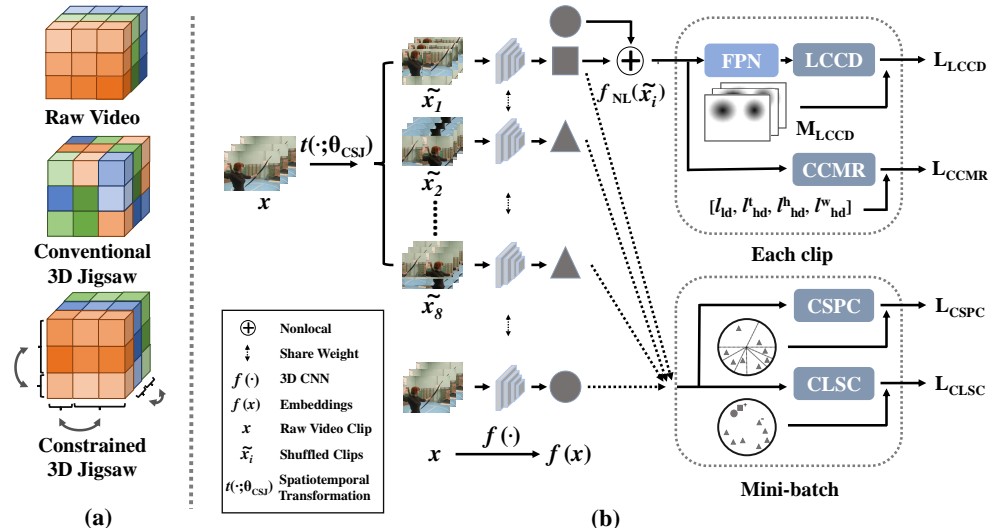

Figure 2: (a) Illustration of our Constrained Spatiotemporal Jigsaw (CSJ) (see Sec. 3.2). (b) The pipeline of our proposed framework for self-supervised video representation learning (see Sec. 3.3). A raw video clip is transformed into 8 shuffled clips with our Constrained Spatiotemporal Jigsaw (CSJ), and a 3D CNN sharing weights extracts the feature representations from them. The model is then trained by solving four self-supervised tasks jointly.

$\{1, 2, \ldots, H\}, w_1, w_2 \in \{1, 2, \ldots, W\}$. If all the jigsaw pieces (smallest video clip unit, *e.g.* a pixel or a 3D pixel block) in $c$ keep the same relative order as they were in $x$ (before being shuffled), we call the cuboid $c$ as a continuous cuboid $c^{\text{cont}}$. The cuboid's volume equals $(t_2 - t_1) \times (h_2 - h_1) \times (w_2 - w_1)$, and the largest continuous cuboid (LCC) $c^{\text{cont}}_{\text{max}}$ is the $c^{\text{cont}}$ with the largest volume.

We introduce two permutation strategies to ensure that the volumes of LCCs are large in relation to the whole video clip volume after our shuffling transformation $t(\cdot; \boldsymbol{\theta}_{\text{CSJ}})$. First, instead of shuffling $x$ in three spatiotemporal dimensions simultaneously, $t(\cdot; \boldsymbol{\theta}_{\text{CSJ}})$ factorizes the permutations into the three spatiotemporal dimensions and then utilizes them sequentially to generate shuffled clips, *e.g.*, in the order of $T, W, H$ and only once. Note that the volume of the generated $\widetilde{x}$ stays the same with different permutation orders (*e.g.*, $TWH$ and $HTW$). Second, we shuffle a group of jigsaw pieces together instead of each piece individually along each dimension. Taking spatial shuffling as an example, if there are 8 pieces per frame (along each of the two spatial dimensions), $\boldsymbol{\theta}_{\text{CSJ}}$ could be represented as the permutation from $\{12345678\}$ to $\{84567123\}$. The longest and the second-longest index ranges are: $[2, 5]$ for coordinates $\{4567\}$, and $[6, 8]$ for coordinates $\{123\}$. With these two permutation strategies, not only do we have large LCCs, but also they are guaranteed to have clearly separable boundaries (see Fig. 2(b)) with surrounding pieces due to the factorized and grouped permutation design. This means that they are easily detectable.

## 3.3 SURROGATE TASKS

Having permutation constraints preserves more spatiotemporal continuity in the shuffled clip and reduces the amount of possible permutations. But exploiting these constraints to make a neural sorting model tractable is still far from trivial. Instead of solving the jigsaw directly, our $P_{\text{CSJ}}$ is thus formulated as four surrogate tasks: Largest Continuous Cuboid Detection (LCCD), Clip Shuffling Pattern Classification (CSPC), Contrastive Learning over Shuffled Clips (CLSC), and Clip Continuity Measure Regression (CCMR). As illustrated in Fig. 2(b), given an unlabeled clip $x$, we first construct a mini-batch of 8 clips $\{\widetilde{x}_1, \widetilde{x}_2, ..., \widetilde{x}_8\}$ by shuffling $x$ with different but related constrained permutations (to be detailed later). These shuffled clips and the raw clip $x$ are then fed into a 3D CNN model $f(\cdot)$ for spatiotemporal representation learning with a non-local operation (Wang et al., 2018):

$$f_{\text{NL}}(\widetilde{x}_i) = \text{NL}(f(\widetilde{x}_i), f(x)), \qquad (1)$$

where $\text{NL}(\cdot, \cdot)$ denotes the non-local operator, and $f(\widetilde{x}_i)$ and $f(x)$ denote the feature map of $\widetilde{x}_i$ and $x$ from the last convolutional layer of $f(\cdot)$, respectively. The resultant feature map $f_{\text{NL}}(\widetilde{x}_i)$ is further passed through a spatial pooling layer followed by a separately fully-connected layer for

each surrogate task. Note that the raw video feature map $f(\boldsymbol{x})$ is used as guidance through the non-local based attention mechanism to help fulfill the tasks. This is similar to humans needing to see the completed jigsaw picture to help solve the puzzle.

Before we detail the four tasks, we first explain how the eight permutations from the same raw clip are generated. First, the factorized and grouped permutations are applied to $\boldsymbol{x}$ to create one shuffled clip. By examining the largest and the second-largest continuous puzzle piece numbers of each dimension ($\{T, H, W\}$), we can easily identify the top-2 largest continuous cuboids (LCCs). Next, by varying the relative order of the top-2 LCCs either in the correct (original) order or the reverse order in each dimension, $2 \times 2 \times 2 = 8$ permutations are obtained. By controlling the group size in permutation, we can make sure that the top-2 LCCs account for a large proportion, saying 80% of the total clip volume. Our four tasks are thus centered around these two LCCs as they largely determine the overall spatiotemporal continuity of the shuffled clip.

The first task **LCCD** is to locate the top-2 LCCs $\{\boldsymbol{c}_{\max}^{\mathrm{cont}}(j) : j = 1, 2\}$ and formulated as a regression problem. Given a ground-truth LCC $\boldsymbol{c}_{\max}^{\mathrm{cont}}(j)$, a Gaussian kernel is applied to its center to depict the possibility of each pixel in $\widetilde{\boldsymbol{x}}$ belonging to the LCC. This leads to a soft mask $M_{\mathrm{LCCD}}^{j}$ with the same size of $\widetilde{\boldsymbol{x}}$: $M_{\mathrm{LCCD}}^{j}$ is all 0 outside the region of $\boldsymbol{c}_{\max}^{\mathrm{cont}}(j)$, and $\exp(-\dfrac{||\boldsymbol{a} - \boldsymbol{a}_{\mathrm{c}}||^2}{2\sigma_g^2})$ inside the region, where $\boldsymbol{a}, \boldsymbol{a}_{\mathrm{c}}$ denote any pixel and the center point, respectively. $\sigma_g$ is the hyper-parameter which is set as 1 empirically. In the training stage, FPN (Lin et al., 2017) is used for multi-level feature fusion. LCCD is optimized using the MSE loss in each point:

$$L_{\mathrm{LCCD}} = \sum_{j \in \{1,2\}} \sum_{\boldsymbol{a} \in \widetilde{\boldsymbol{x}}} \mathrm{MSE}(M_{\mathrm{LCCD}}^{j}(\boldsymbol{a}), M_{\mathrm{LCCD}}^{j}(\boldsymbol{a})^{'}), \qquad (2)$$

where $\mathrm{MSE}(\cdot, \cdot)$ denotes the MSE loss function, and $M_{\mathrm{LCCD}}^{j}(\boldsymbol{a})^{'}$ is the prediction of each pixel $\boldsymbol{a}$.

**CSPC** is designed to recognize the shuffling pattern of a shuffled clip. As mentioned early, the eight shuffled clips in each mini-batch are created from the same raw clip and differ only in the relative order of the top-2 LCCs along each of the three dimensions. There are thus eight permutations depending on the order (correct or reverse) in each dimension. Based on this understanding, CSPC is formulated as a multi-class classification task to recognize each shuffled clip into one of these eight classes, which is optimized using the Cross-Entropy (CE) loss:

$$L_{\mathrm{CSPC}} = \sum_{i \in \{0,1,\dots,7\}} \mathrm{CE}(l_{\mathrm{CSPC}}[i], l_{\mathrm{CSPC}}^{'}[i]), \qquad (3)$$

where $\mathrm{CE}(\cdot, \cdot)$ denotes the CE loss function and $l_{\mathrm{CSPC}}^{'}[i]$ is the predicted class label of $i$-th sample (shuffled clip) in each mini-batch.

The two tasks above emphasize on local spatiotemporal continuity understanding. In contrast, **CLSC** leverages the contrastive loss to encourage global continuity understanding. In particular, since the top-2 LCCs dominate the volume of a clip, it is safe to assume that if their relative order is correct in all three dimensions, the shuffled clip largely preserve continuity compared to the original clip, while all other 7 permutations feature large discontinuity in at least one dimension. We thus form a contrastive learning task with the original video $\boldsymbol{x}$ and the most continuous shuffled video $\widetilde{\boldsymbol{x}}_i$ as a positive pair, and $\boldsymbol{x}$ and the rest $\widetilde{\boldsymbol{x}}_j$ ($j \neq i$) as negative pairs. CLSC is optimized using the Noise Contrastive Estimation (NCE) (Tian et al., 2020) loss:

$$L_{\mathrm{CLSC}} = -\log \frac{\exp(\mathrm{sim}(f(\boldsymbol{x}), f(\widetilde{\boldsymbol{x}}_i))/\tau)}{\exp(\mathrm{sim}(f(\boldsymbol{x}), f(\widetilde{\boldsymbol{x}}_i))/\tau) + \sum_j \exp(\mathrm{sim}(f(\boldsymbol{x}), f(\widetilde{\boldsymbol{x}}_j))/\tau)}, \qquad (4)$$

where $\mathrm{sim}(\cdot, \cdot)$ is defined by the dot product: $f(\boldsymbol{x})^{\top} f(\widetilde{\boldsymbol{x}}_i)$, and $\tau$ is the temperature hyper-parameter. Note that the non-local operator is not used in CLSC.

**CCMR** is similar to CLSC in that it also enforces global continuity understanding, but differs in that it is a regression task aimed at predicting a global continuity measure. We consider two such measures. Since the total size of the top-2 LCCs $\{\boldsymbol{c}_{\max}^{\mathrm{cont}}(j) : j = 1, 2\}$ is a good indicator of how continuous a shuffle video clip is, the first measure $l_{ld}$ directly measures the relative total size of the top-2 LCCs: $l_{ld} = \dfrac{\mathrm{v}(\boldsymbol{c}_{\max}^{\mathrm{cont}}(1)) + \mathrm{v}(\boldsymbol{c}_{\max}^{\mathrm{cont}}(2))}{\mathrm{v}(\widetilde{\boldsymbol{x}})}$, where $\mathrm{v}(\cdot)$ represents the volume of a clip/cuboid.

The second measure $l_{\text{hd}}^{\text{t/h/w}}$ examines the shuffling degree of $\widetilde{\boldsymbol{x}}$ in each dimension, computed as the normalized hamming distance: $\dfrac{\text{hamming}(\widetilde{\boldsymbol{x}})}{N_c(N_c-1)/2}$, where $\text{hamming}(\cdot)$ denotes the hamming distance in each dimension between the original piece sequence and the permuted one, and $N_c$ represents the number of pieces in each dimension so that $N_c(N_c-1)/2$ indicates the maximum possible hamming distance in the dimension. CCMR is optimized using the Mean Squared Error (MSE) loss:

$$L_{\text{CCMR}} = \text{MSE}([l_{\text{ld}}, l_{\text{hd}}^{\text{t}}, l_{\text{hd}}^{\text{h}}, l_{\text{hd}}^{\text{w}}], [l_{\text{ld}}^{'}, l_{\text{hd}}^{\text{t}'}, l_{\text{hd}}^{\text{h}'}, l_{\text{hd}}^{\text{w}'}]), \tag{5}$$

where $l_{\text{ld}}^{'}, l_{\text{hd}}^{\text{t}'}, l_{\text{hd}}^{\text{h}'}, l_{\text{hd}}^{\text{w}'}$ are the prediction of the model.

## 3.4 OVERALL LEARNING OBJECTIVE

Our entire CSJ framework is optimized end-to-end with the learning objective defined as:

$$L = \sigma_1 L_{\text{LCCD}} + \sigma_2 L_{\text{CSPC}} + \sigma_3 L_{\text{CLSC}} + \sigma_4 L_{\text{CCMR}}, \tag{6}$$

where $\sigma_1, \sigma_2, \sigma_3, \sigma_4$ denote the weights for the four losses. We deploy the adaptive weighting mechanism (Kendall et al., 2018) to weight these tasks, and thus there is no free hyper-parameters to tune. We also adopt curriculum learning (Bengio et al., 2009; Korbar et al., 2018) to train our network by shuffling clips from easy to hard. More details are presented in Appendix. A.1 and A.2.

## 4 EXPERIMENTS

### 4.1 DATASETS AND SETTINGS

We select three benchmark datasets for performance evaluation: UCF101 (Soomro et al., 2012), HMDB51 (Kuehne et al., 2011), and Kinetics-400 (K400) (Kay et al., 2017), containing 13K/7K/306K video clips from 101/51/400 action classes, respectively. In the self-supervised pre-training stage, we utilize the first training split of UCF101/HMDB51 and the training split of K400 without using their labels. As in Han et al. (2020), we adopt R2D3D as the backbone network, which is modified from R3D (Hara et al., 2018) with fewer parameters. By fine-tuning the pre-trained model, we can evaluate the SSL performance on a downstream task (*i.e.*, action classification). Following Han et al. (2019); He et al. (2020), two evaluation protocols are used: comparisons against state-of-the-arts follow the more popular fully fine-tuning evaluation protocol, but ablation analysis takes both the linear evaluation and fully fine-tuning protocols. For the experiments on supervised learning, we report top-1 accuracy on the first test split of UCF101/HMDB51 as the standard (Han et al., 2020). More details of the datasets are provided in Appendix B.

### 4.2 IMPLEMENTATION DETAILS

Raw videos in these datasets are decoded at a frame rate of 24-30 fps. From each raw video, we start from a randomly selected frame index and sample a consecutive 16-frame video clip with a temporal stride of 4. For data augmentation, we first resize the video frames to $128 \times 171$ pixels, from which we extract random crops of size $112 \times 112$ pixels. We also apply random horizontal flipping and random color jittering to the video frames during training. We exploit *only the raw RGB video frames* as input, and do not leverage optical flow or other auxiliary signals for self-supervised pre-training. We adopt the Adam optimizer with a weight decay of $10^{-3}$ and a batch size of 8 per GPU (with a total of 32 GPUs). We deploy cosine annealing learning rate with an initial value of $10^{-4}$ and 100 epochs. The jigsaw puzzle piece sizes of $\{T, H, W\}$ dimensions are set as $1, 4, 4$, respectively. A $16 \times 112 \times 112$ video clip thus contains $16 \times 28 \times 28$ pieces. We set the temperature hyper-parameter $\tau$ to 0.07. A dropout of 0.5 is applied to the final layer of each task. More implementation details of the fine-tuning and test evaluation stages can be found in Appendix B.

### 4.3 MAIN RESULTS

**Comparison in Action Recognition** A standard way to evaluate a self-supervised video representation learning model is to use it to initialize an action recognition model on a small dataset. Specifically, after self-supervised pre-training on UCF101/HMDB51/K400, we exploit the learned backbone for fully fine-tuning on UCF101 and HMDB51, following Han et al. (2020); Wang et al. (2020).

Table 1: Comparison to the state-of-the-art on UCF101(U) and HMDB51(H). All models are pre-trained with the RGB modality only. †: Methods with temporal-only transformations. ‡: Methods with both spatial and temporal transformations. ∗: Methods that leverage spatiotemporal representations. HT: HowTo100M. The underline represents the second-best result.

| Methods | Backbone | Pre-trained Datasets | Input size (T*H) | UCF101 | HMDB51 |
|---|---|---|---|---|---|
| CMC (ECCV'20) (Tian et al., 2020) | C2D | U | – | 55.3 | – |
| Skip-Clip† (El-Nouby et al., 2019) | R3D-18 | U/H | 16*112 | 64.4 | – |
| VCOP† (CVPR'19) (Xu et al., 2019) | R3D-18 | U/H | 16*112 | 64.9 | 29.5 |
| VCP† (AAAI'20) (Luo et al., 2020b) | R3D-18 | U/H | 16*112 | 66.0 | 31.5 |
| PRP† (CVPR'20) (Yao et al., 2020) | R3D-18 | U/H | 16*112 | 66.5 | 29.7 |
| MemDPC* (ECCV'20) (Han et al., 2020) | R2D3D-18 | U/H | 40*128 | 69.2 | – |
| **CSJ*** **(ours)** | R2D3D-18 | U/H | 16*112 | **70.4** | **36.0** |
| Video-Jigsaw‡ (WACV'19) (Ahsan et al., 2019) | C2D | K400 | 25*224 | 55.4 | 27.0 |
| Statisics* (CVPR'19) (Wang et al., 2019) | C3D | K400 | 16*112 | 61.2 | 33.4 |
| ST-Puzzle‡ (AAAI'19) (Kim et al., 2019) | R3D-18 | K400 | 16*112 | 63.9 | 33.7 |
| DPC*(ICCVW'19) (Han et al., 2019) | R2D3D-18 | K400 | 40*128 | 68.2 | 34.5 |
| SpeedNet† (CVPR'20) (Benaim et al., 2020) | I3D | K400 | 16*224 | 66.7 | 43.7 |
| VIE* (CVPR'20) (Zhuang et al., 2020) | R3D-18 | K400 | 16*112 | 75.5 | 44.6 |
| Pace† (ECCV'20) (Wang et al., 2020) | R(2+1)D-18 | K400 | 16*112 | **77.1** | 36.6 |
| **CSJ*** **(ours)** | R2D3D-18 | K400 | 16*112 | 76.2 | **46.7** |
| MemDPC* (ECCV'20) (Han et al., 2020) | R2D3D-34 | K400 | 40*224 | 78.1 | 41.2 |
| CBT (Sun et al., 2019) | S3D | K600+HT | – | **79.5** | 44.6 |
| **CSJ*** **(ours)** | R2D3D-34 | K400 | 16*224 | **79.5** | **50.9** |
| **Upper Bound**: Fully-Supervised | R3D-34 | K400 | 16*224 | 87.7 | 59.1 |

Table 2: Comparison with state-of-the-art self-supervised learning methods for nearest neighbor video retrieval (top-$k$ recall) on UCF101. The underline represents the second-best result.

| Methods | Backbone | Top1 | Top5 | Top10 | Top20 | Top50 |
|---|---|---|---|---|---|---|
| VCOP (CVPR'19) (Xu et al., 2019) | R3D-18 | 14.1 | 30.3 | 40.0 | 51.1 | 66.5 |
| VCP (AAAI'20) (Luo et al., 2020b) | R3D-18 | 18.6 | 33.6 | 42.5 | 53.5 | 68.1 |
| SpeedNet (CVPR'20) (Benaim et al., 2020) | S3D-G | 13.1 | 28.1 | 37.5 | 49.5 | 65.0 |
| PRP (CVPR'20) (Yao et al., 2020) | R3D-18 | **22.8** | 38.5 | 46.7 | 55.2 | 69.1 |
| Pace (ECCV'20) (Wang et al., 2020) | R3D-18 | 19.9 | 36.2 | 46.1 | 55.6 | 69.2 |
| ERUV (Luo et al., 2020a) | R3D-18 | 21.4 | 35.2 | 43.8 | 53.1 | 68.3 |
| MemDPC (ECCV'20) (Han et al., 2020) | R2D3D-18 | 20.2 | 40.4 | 52.4 | 64.7 | – |
| CSJ (ours) | R2D3D-18 | 21.5 | **40.5** | **53.2** | **64.9** | **70.0** |

We consider one baseline: fully-supervised learning with pre-training on K400. Note that this baseline is commonly regarded as the upper bound of self-supervised representation learning (Alwassel et al., 2019). From Table 1, we have the following observations: (1) Our CSJ achieves state-of-the-art performance on both UCF101 and HMDB51. Particularly, with the backbone R2D3D-18 that is weaker than R(2+1)D-18, our CSJ performs comparably w.r.t. Pace on UCF101 but achieves a 10% improvement over Pace on HMDB51. (2) By exploiting spatiotemporal transformations for self-supervised representation learning, our CSJ beats either methods with only temporal transformations (†) or methods with both spatial and temporal transformations (‡), as well as those learning spatiotemporal representations (∗) via only contrastive learning (w./o. spatiotemporal transformations). (3) Our CSJ also outperforms CBT (Sun et al., 2019), which used ten-times more massive datasets (K600 (Carreira et al., 2018) + Howto100M (Miech et al., 2019)) and multiple modalities (RGB+Audio). (4) Our CSJ is the closest to the fully-supervised one (upper bound), validating its effectiveness in self-supervised video representation learning.

**Comparison in Video Retrieval**   We evaluate our CSJ method in the video retrieval task. Following Xu et al. (2019), we extract each video clips' embeddings with the pre-training model and use each clip in the test set to query the $k$ nearest clips in the training set. The comparative results in Table 2 show that our method outperforms all other self-supervised methods and achieves new state-of-the-art in video retrieval on UCF101. Particularly, our method beats the latest competitor

Table 3: Evaluation of pre-training tasks with the backbone R2D3D-18 under linear probe and fully fine-tuning protocols on UCF101. AW: Adaptive Weighting. CL: Curriculum Learning.

| Tasks | Linear Probe | Fully Fine-tuning |
|---|---|---|
| Random Initialization | 8.3 | 63.6 |
| LCCD | 21.8 | 67.8 |
| CSPC | 22.6 | 68.1 |
| CLSC | 18.9 | 68.1 |
| CCMR | 22.7 | 68.1 |
| CCMR+CSPC | 24.7 | 69.2 |
| CCMR+CSPC+CLSC | 25.5 | 69.3 |
| CCMR+CSPC+CLSC+LCCD | 27.9 | 69.5 |
| CCMR+CSPC+CLSC+LCCD+AW | 28.2 | 70.0 |
| CCMR+CSPC+CLSC+LCCD+AW+CL | **28.5** | **70.4** |

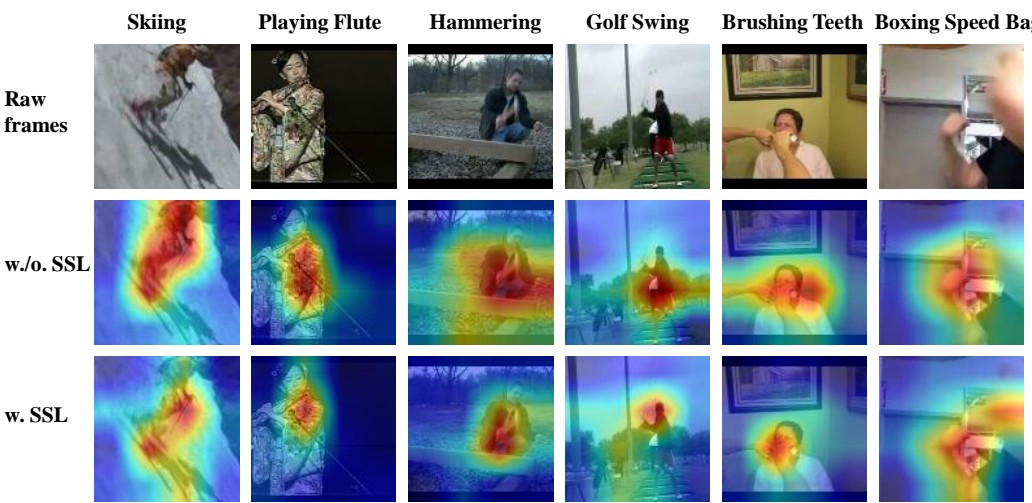

Figure 3: Attention visualization of the last feature maps from the fine-tuned models on UCF101. The first row denotes the raw frames from videos, and the last two rows correspond to fine-tuning from random initialization and our self-supervised pre-trained model, respectively.

PRP (Yao et al., 2020) on four out of five metrics. This indicates that our proposed CSJ is also effective for video representation learning in video retrieval.

## 4.4 FURTHER EVALUATIONS

**Ablation Study** We conduct ablative experiments to validate the effectiveness of four CSJ surrogate tasks and two additional learning strategies. From Table 3, we can observe that: (1) Self-supervised learning with each of the four tasks shows better generalization than fine-tuning the network from scratch (random initialization). (2) By training over all the four tasks jointly, we can achieve large performance gains (see '+LCCD' vs. 'CCMR'). (3) Each additional learning strategy (*i.e.*, adaptive weighting or curriculum learning) leads to a small boost to the performance by 0.3-0.5%. (4) Our full model achieves a remarkable classification accuracy of 70.4%, demonstrating the effectiveness of our proposed CSJ with only the RGB video stream (without additional optical flow, audio, or text modalities). More ablative analysis can be found in Appendix D.

**Visualization of Attention Maps** Fig. 3 visualizes the attention map of the last feature maps from two models fine-tuned on UCF101 with or without adopting our self-supervised pre-training. Since each frame's attention map involves four adjacent frames, it actually contains spatiotemporal semantic features. We can see that our self-supervised pre-training with CSJ indeed helps to better capture meaningful spatiotemporal information and thus recognize the action categories more correctly.

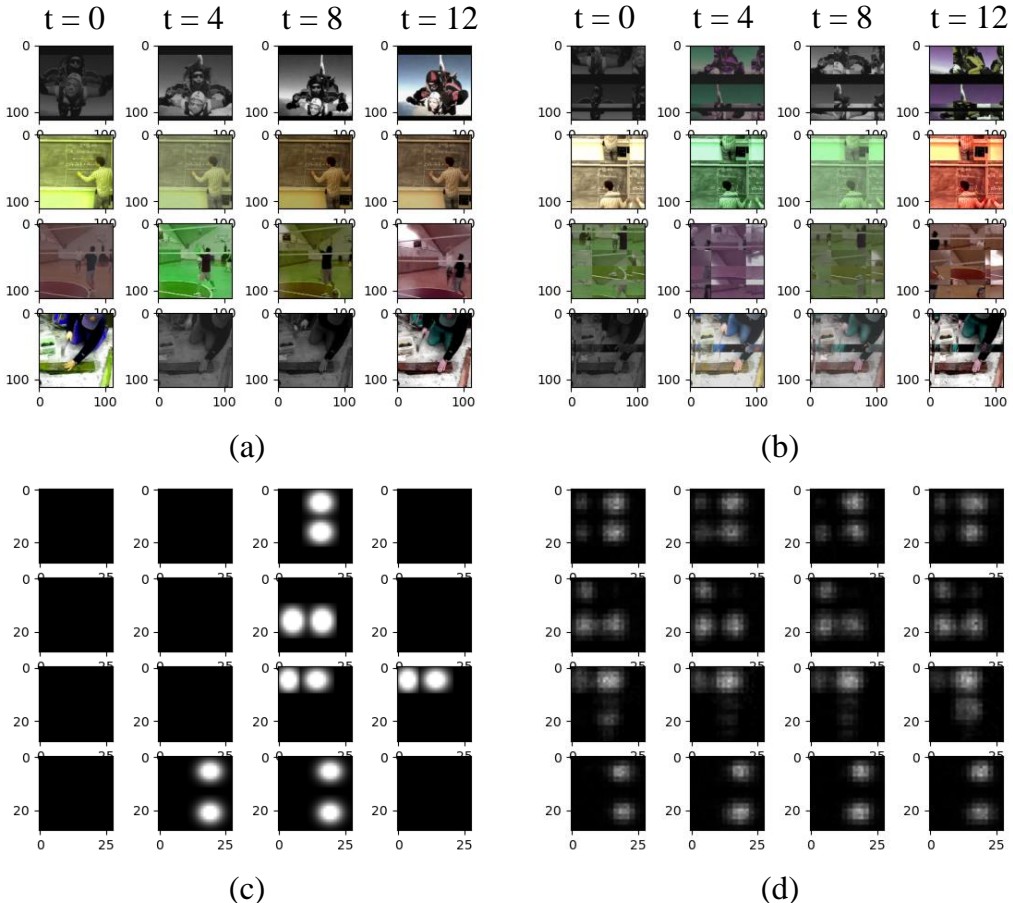

Figure 4: Visualization of the LCCD predictions from the pre-trained models. Each row denotes the frames at time stamp = (0, 4, 8, 12) from one video clip. (a) raw frames (with color jittering); (b) shuffled frames; (c) the ground truth of LCCD; (d) network's prediction.

**Visualization of LCCD Predictions** We also demonstrate the visualization of the LCCD predictions from the pre-trained models in Fig. 4. We can observe that solving the LCCD task indeed enables the model to learn the locations of LCCs and understand spatiotemporal continuity, which is a key step towards video content understanding.

## 5 CONCLUSION

We have introduced a novel self-supervised video representation learning method named Constrained Spatiotemporal Jigsaw (CSJ). By introducing constrained permutations, our proposed CSJ is the first to leverage spatiotemporal jigsaw in self-supervised video representation learning. We also propose four surrogate tasks based on our constrained spatiotemporal jigsaws. They are designed to encourage a video representation model to understand the spatiotemporal continuity, a key building block towards video content analysis. Extensive experiments were carried out to validate the effectiveness of each of the four CSJ tasks and also show that our approach achieves the state-of-the-art on two downstream tasks across various benchmarks.

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

## A  ADDITIONAL LEARNING STRATEGIES

### A.1  ADAPTIVE WEIGHT

Formally, our CSJ has two continuous outputs $y_1, y_4$ from LCCD and CCMR, and two discrete outputs $y_2, y_3$ from CSPC and CLSC, modeled with Gaussian likelihoods and softmax likelihoods, respectively. The joint loss for these four tasks $L(\mathbf{W}, \sigma_1, \sigma_2, \sigma_3, \sigma_4)$ is:

$$
\begin{aligned}
&L(\mathbf{W}, \sigma_1, \sigma_2, \sigma_3, \sigma_4) \\
&= -\log \mathcal{N}(\mathbf{y}_1; \mathbf{f^W}(\mathbf{x}), \sigma_1^2) \cdot -\log \mathcal{N}(\mathbf{y}_4; \mathbf{f^W}(\mathbf{x}), \sigma_4^2) \\
&\quad \cdot \text{softmax}(\mathbf{y}_2 = c; \mathbf{f^W}(\mathbf{x}), \sigma_2) \cdot \text{softmax}(\mathbf{y}_3 = c; \mathbf{f^W}(\mathbf{x}), \sigma_3) \\
&= \frac{1}{2\sigma_1^2} ||\mathbf{y}_1 - \mathbf{f^W}(\mathbf{x})||^2 + \log \sigma_1 - \frac{1}{2\sigma_4^2} ||\mathbf{y}_4 - \mathbf{f^W}(\mathbf{x})||^2 + \log \sigma_4 \qquad (7) \\
&\quad - \log p(\mathbf{y}_2 | \mathbf{f^W}(\mathbf{x}), \sigma_2) - \log p(\mathbf{y}_3 | \mathbf{f^W}(\mathbf{x}), \sigma_3) \\
&\approx \frac{1}{2\sigma_1^2} L_1(\mathbf{W}) + \frac{1}{\sigma_2^2} L_2(\mathbf{W}) + \frac{1}{\sigma_3^2} L_3(\mathbf{W}) + \frac{1}{2\sigma_4^2} L_4(\mathbf{W}) \\
&\quad + \log \sigma_1 + \log \sigma_2 + \log \sigma_3 + \log \sigma_4,
\end{aligned}
$$

where $\sigma$ is the weight factor that can be automatically learned from the network, and the log likelihood for the output $y$ is defined as:

$$
\log p(\mathbf{y} = c | \mathbf{f^W}(\mathbf{x}), \sigma) = \frac{1}{\sigma^2} f_c^{\mathbf{W}}(\mathbf{x}) - \log \sum_{c'} \exp(\frac{1}{\sigma^2} f_{c'}^{\mathbf{W}}(\mathbf{x})). \qquad (8)
$$

### A.2  CURRICULUM LEARNING

We adopt curriculum learning (Korbar et al., 2018) to train our network by shuffling clips from easy to hard. Let $d$ be the shuffle degree of a shuffled clip $\widetilde{x}$, representing the number of continuous cuboids in each dimension. We gradually increase $d$ from 3 to 5 during the training phase to produce more permuted clips. Note that when the video content is ambiguous in one dimension, *e.g.*, a static video clip inflated from an image, there is no temporal variance to learn the transformation. Kim et al. (2019); Noroozi & Favaro (2016) also mentioned this problem as similar-looking ambiguity. To solve this problem, we calculate the variance on each dimension and set a threshold. If the variance is lower than the threshold, we decrease $d$ from 3 to 1 so that the pieces are not shuffled in the corresponding dimension.

## B  DATASETS AND IMPLEMENTATION

### B.1  DETAILS OF DATASETS

**UCF101** (Soomro et al., 2012) is a widely-used dataset in the action recognition task, which contains 13,320 videos with 101 action classes. The dataset is divided into three training/testing splits. In this paper, following prior works (Wang et al., 2020; Han et al., 2020), we use the first training split as the pre-training dataset and the first testing split for evaluation.

**HMDB51** (Kuehne et al., 2011) is a relatively small action recognition dataset, consisting of 6,766 videos with 51 categories. It is also divided into three training/testing splits. Following Wang et al. (2020); Han et al. (2020), we use the first training split as the pre-training dataset and the first testing split for evaluation.

**Kinetics-400** (K400) (Kay et al., 2017) is a very large action recognition dataset consisting of 400 human action classes and around 306k videos. In this work, we use the training split of K400 as the pre-training dataset.

### B.2  IMPLEMENTATION DETAILS

In the fine-tuning stage, weights of convolutional layers are initialized with self-supervised pre-training, but weights of fully-connected layers are randomly initialized. The whole network is then trained with the cross-entropy loss. The pre-processing and training strategies are the same as in the

Table 4: The structure of the encoding function $f(\cdot)$. R2D3D-18 is used as an example.

| stage | detail | output size $T \times HW \times C$ |
|---|---|---|
| input data | - | $16 \times 112^2 \times 3$ |
| conv$_1$ | $1 \times 7^2, 64$ 
 stride $1, 2^2$ | $16 \times 56^2 \times 64$ |
| pool$_1$ | $1 \times 3^2, 64$ 
 stride $1, 2^2$ | $16 \times 28^2 \times 64$ |
| res$_2$ | $\begin{bmatrix} 1 \times 3^2, 64 \\ 1 \times 3^2, 64 \end{bmatrix} \times 2$ | $16 \times 28^2 \times 64$ |
| res$_3$ | $\begin{bmatrix} 1 \times 3^2, 128 \\ 1 \times 3^2, 128 \end{bmatrix} \times 2$ | $16 \times 14^2 \times 128$ |
| res$_4$ | $\begin{bmatrix} 3 \times 3^2, 256 \\ 3 \times 3^2, 256 \end{bmatrix} \times 2$ | $8 \times 7^2 \times 256$ |
| res$_5$ | $\begin{bmatrix} 3 \times 3^2, 512 \\ 3 \times 3^2, 512 \end{bmatrix} \times 2$ | $4 \times 4^2 \times 512$ |
| Avgpool | $4 \times 4^2, 512$ 
 stride $1, 1^2$ | $1 \times 1^2 \times 512$ |

self-supervised pre-training stage, except that the total epochs are 300 and the initial learning rate is $10^{-3}$. We use a batch size of 64 per GPU and a total of 8 GPUs for fine-tuning.

We follow the standard evaluation protocol (Han et al., 2020) during inference and use ten-crop to take the same sequence length as training from the video. The predicted label of each video is calculated by averaging the softmax probabilities of all clips in the video.

## C  NETWORK ARCHITECTURE

We deploy the same network backbone R2D3D as Han et al. (2019; 2020), which is a 3D-ResNet (R3D) similar to Hara et al. (2018). The only difference between R2D3D and R3D lies in that: R2D3D keeps the first two residual blocks as 2D convolutional blocks while R3D uses 3D blocks. Therefore, the modified R2D3D has fewer parameters (only the last two blocks are 3D convolutions). We present the CNN structure of R2D3D in Table 4.

## D  ADDITIONAL ABLATION STUDIES

Table 5: Evaluation of pre-training tasks under different designs of LCCD on UCF101.

| Methods | LCCS | LCCD+$M_{\text{LCCS}}$ | LCCD + L1 | LCCD + MSE |
|---|---|---|---|---|
| Top-1 Acc | 66.5 | 64.0 | 66.5 | 67.8 |

### D.1  LCCD

Instead of predicting center points using the detection method, we also design a segmentation method – largest continuous cuboid segmentation (LCCS) to predicts the location of top-2 LCCs $\{c_{\max}^{\text{cont}}(j) : j = 1, 2\}$. The difference between LCCD and LCCS lies in that: LCCS is formulated as a segmentation task to discriminate whether a pixel is in the region of $c_{\max}^{\text{cont}}(j)$. Concretely, LCCS predicts a binary mask $M_{\text{LCCS}}^j$ where only points in the region of $\{c_{\max}^{\text{cont}}(j)$ are set to be 1, otherwise 0. As a result, LCCS is optimized using the Cross Entropy (CE) loss at each point:

$$L_{\text{LCCS}} = \sum_{j \in \{1,2\}} \sum_{\boldsymbol{a} \in \widetilde{\boldsymbol{x}}} \text{CE}(M_{\text{LCCS}}^j(\boldsymbol{a}), M_{\text{LCCS}}^j(\boldsymbol{a})^{'}), \tag{9}$$

where $\text{CE}(\cdot, \cdot)$ denotes the CE loss function, and $M_{\text{LCCS}}^j(\boldsymbol{a})^{'}$ is the predicted class of pixel $\boldsymbol{a}$.

We report the performance of four different designs of LCCD in Table 5: (1) LCCS: LCCS is used instead of LCCD. (2) LCCD+$M_{\text{LCCS}}$: The Gaussian mask $M_{\text{LCCD}}$ is substituted by the binary mask $M_{\text{LCCS}}$, but the LCCD task is optimized using the MSE loss. (3) LCCD + L1: The LCCD task is

optimized by the L1 loss. (4) LCCD + MSE: The LCCD task is optimized by the MSE loss. From Table 5, it can be seen that the segmentation task also helps self-supervised representation learning but doesn't perform as well as LCCD. Also, under the three different settings of LCCD, the MSE loss with the Gaussian map performs the best.

Table 6: Evaluation of different temperature $\tau$ for CLSC on UCF101.

| Methods | $\tau = 1$ | $\tau = 0.1$ | $\tau = 0.07$ |
|---------|------------|--------------|---------------|
| Top-1 Acc | 60.9 | 66.3 | 68.1 |

### D.2 CLSC

Table 6 above shows the accuracies obtained with different temperatures $\tau$ used in contrastive learning. We can observe that: (1) When $\tau$ is in the range $1 \sim 0.07$, the accuracy increases with smaller $\tau$. (2) When $\tau$ is large (*e.g.*, 1), the accuracy drops considerably. In this work, $\tau$ is set to 0.0.

Table 7: Evaluation of different designs of CSPC on UCF101.

| Methods | 2 Categories | 4 Categories | 8 Categories |
|---------|--------------|--------------|--------------|
| Top-1 Acc | 67.0 | 68.0 | 68.1 |

### D.3 CSPC

In addition to our CSPC with 8 pattern categories (see Sec. 3.3), we consider another two designs: (1) 2 Categories: the shuffled clip is discriminated by whether it has the same relative order of the top-2 LCCs as the raw clip. It is almost the same as CLSC but is optimized by the CE loss. (2) 4 Categories: the shuffled clip is discriminated by how it differs from the raw clip: non-difference, spatial-only difference, temporal-only difference, spatiotemporal difference. From Table 7, we can see that CSPC with 8 categories outperforms the other two designs. These results support our motivation for leveraging spatiotemporal transformations.

Table 8: Evaluation of different designs of CCMR on UCF101.

| Methods | ld | hd | ld+hd |
|---------|-----|-----|-------|
| Top-1 Acc | 66.9 | 67.2 | 68.1 |

### D.4 CCMR

We report the performance of three different designs of CCMR: (1) ld: the learning degree $l_{ld}$ is used as supervision, which only contains volume information. (2) hd: the hamming distances $l_{hd}^t, l_{hd}^h, l_{hd}^w$ are used, which contain only the relative order information. (3) ld + hd: both ld and hd are used as supervision. From Table 8, we can see that: First, both ld and hd help the model to learn continuous characteristics during pre-training, and hd outperforms ld by a small margin. Second, our CCMR learns the best representation by combining ld and hd.

### D.5 RESULTS OF DIRECTLY SOLVING CSJ

We also demonstrate the results of solving the CSJ task directly in Table 9. We randomly shuffle video clips into $4 \times 4 \times 4$ jigsaw puzzles. To recognize the correct permutation, the model solve a $(4! \times 4! \times 4!)$-way classification task in the pre-training stage. We compare the CSJ task with the joint LCCD+CCMR task under the same setting for fair comparison. Linear evaluation is adopted to show the effectiveness of different tasks. We can observe from the table that solving LCCD+CCMR jointly is more effective than solving CSJ directly.

## E  TEMPORAL ACTION SEGMENTATION

To show the effectiveness of our CSJ for solving new downstream tasks, we apply the pretrained model obtained by our CSJ to temporal action segmentation, which is more challenging than the

Table 9: Evaluation of pre-training tasks with the backbone R2D3D-18 under the linear evaluation protocol on UCF101. For computation efficiency, CSJ is only defined on $4 \times 4 \times 4$ cells.

| Random Initialization | CSJ | LCCD+CCMR ($4 \times 4 \times 4$) | LCCD+CCMR ($16 \times 28 \times 28$) |
|---|---|---|---|
| 8.3 | 13.8 | 18.2 | 23.1 |

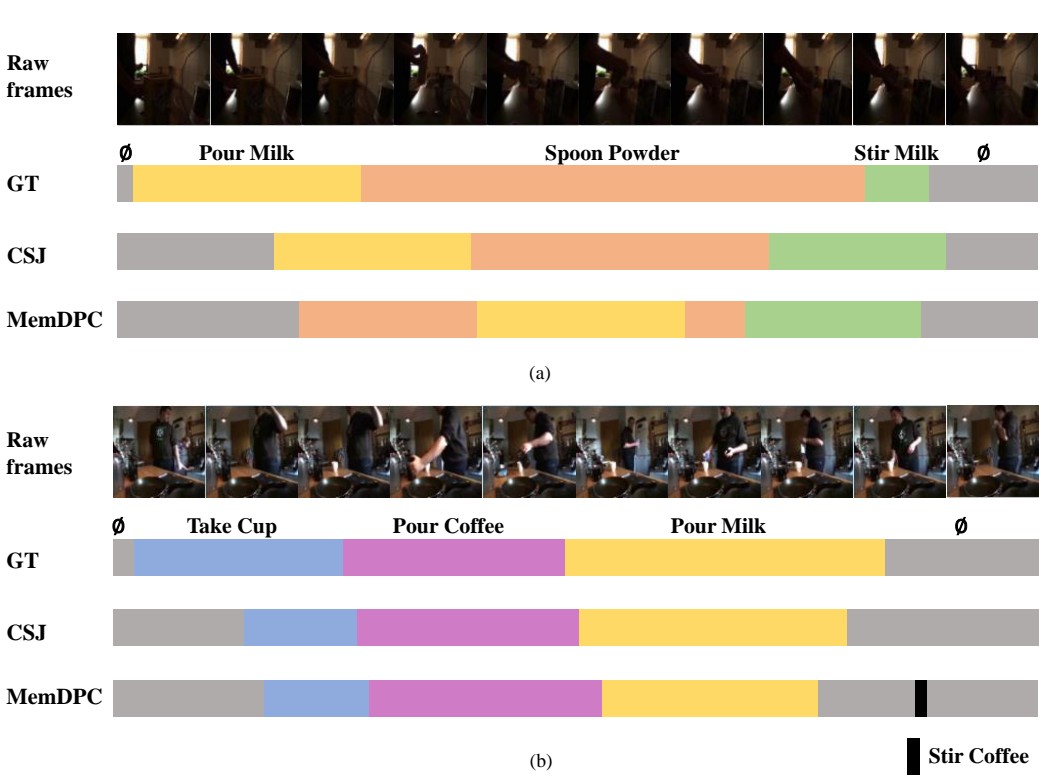

Figure 5: Qualitative results for the temporal action segmentation task on the Breakfast dataset. Note that the notation $\emptyset$ denotes an unannotated segment in the ground truth.

conventional action recognition and retrieval tasks. Specifically, we choose to compare our CSJ model with the latest competitor MemDPC (Han et al., 2020) on the Breakfast dataset (Kuehne et al., 2014). For fair comparison, our CSJ model and the MemDPC model adopt the same R2D3D-34 backbone. Due the time constraint, from the original Breakfast dataset, we only use a small subset of 200 long videos as the training set for fine-tuning, and select a few long videos for the test. For temporal action segmentation, we follow the overall framework of MS-TCN (Abu Farha & Gall, 2019), but changes its backbone to R2D3D-34 pretrained by our CSJ or MemDPC.

We present the qualitative results on two test videos in Fig. 5. We can clearly observe that our CSJ outperforms MemDPC on both test videos. Particularly, the predictions of our CSJ are much closer to the ground truth, but MemDPC tends to produce unwanted segments for temporal action segmentation: it wrongly recognizes the segment (color in yellow) in the middle part of the first video as 'Pour Milk', and the segment (color in black) in the last part of the second video as 'Stir Coffee'. In conclusion, as compared to the latest SSVRL method MemDPC, our CSJ can learn more robust features for temporal action segmentation due to its 'true' spatiotemporal jigsaw understanding.

