# OpenReview forum: "Self-Supervised Video Representation Learning with Constrained Spatiotemporal Jigsaw"
_ICLR.cc/2021/Conference — Reject_

### Official Review · AnonReviewer2 · 2020-10-19
**Interesting work, good improvement by the proposed components, but need more analyses and explanations on the proposed method**

**Rating:** 7
**Confidence:** 4

**Review:**

#### Summary
In this paper, the authors extend the self-supervised 2D jigsaw puzzle solving idea to 3D for self-supervised video representation learning. To make the 3D jigsaw puzzle problem tractable, they propose a two-fold idea. First, they constrain the 3D jigsaw puzzle solution space by factorizing the permutations into time, x, and y dimensions and by grouping pieces. Second, since the constrained 3D jigsaw is still intractable, they propose four surrogate tasks of the 3D jigsaw: 1) LLCD (detecting largest continuous cuboid), 2) CSPC (3D permutation pattern classification), 3) CLSC (contrastive learning over permuted clips), 4) CCMR (measuring the global continuity of the permuted clips)
They evaluate their method's efficacy on the public benchmarks by following linear/finetuning self-supervised learning evaluation protocols.

#### Strengths
I like the idea of solving a 3D jigsaw puzzle as a pretext spatio-temporal learning task. By learning to solve the 3D jigsaw puzzle, the learned representations could be discriminative for the downstream tasks. Solving the jigsaw puzzle as a pretext task for representation learning is already explored and shown to be effective both in 2D spatial for images [Noroozi & Favaro, ECCV 2016] and 1D temporal dimension for videos [Xu et al., CVPR2019]. Nevertheless, due to the problem's intractability, there is no such prior work on solving a jigsaw puzzle for video representation learning. Therefore, I think this work is valuable as the authors make the problem tractable, and they show the efficacy of the 3D jigsaw puzzle solving.

#### Weaknesses and suggestions
However, I have several concerns about the work.
* I do not understand how the puzzle pieces are grouped exactly. The authors show an example of grouped permutation: {12345678} -> {84567123}. It is confusing to me. It seems that the groups are {123}, {4567}, {8} from the original sequence. However, how do we make these groups? Is the group sizes always 1,3,4 for length-8 sequences? I suggest the authors provide more details on how they group the pieces.
* Artificial patterns in the shuffled clips might be problematic. In contrast to the 2D jigsaw [Noroozi & Favaro, ECCV 2016] and 1D jigsaw [Xu et al., CVPR2019], the backbone encoder in this work takes the shuffled clips with artificial patterns (see the Fig. 1(c). There are vertical and horizontal lines). It is unlikely to see these artificial patterns in the downstream tasks. There is a training-testing mismatch. Finetuning might fix the problem, but it is not guaranteed. I want to listen to the authors' opinions on this issue.
* Missing ablation experiments and analysis. I list the missing analyses below.
  1. Why the number of largest continuous cuboids are two? What happens if it is one, three, or four?
  2. They use non-local operation between the permuted and the original features to guide the surrogate tasks. It would be informative to show the performance when we remove this part. Also, I am not quite sure why FPN is used only for LCCD. What happens if we do not use FPN for LCCD?
  3. Analysis of the correlation between surrogate task performance and the downstream task performance.

 I will increase my rating if the majority of concerns are resolved.

#### Minor comments
For me, Table 3 is a bit hard to parse.

---

> ### Author Response · Authors · 2020-11-14
> **Response to AnonReviewer2**
>
> We’d like to thank the reviewer for the constructive comments and suggestions.
>
> **Q1. I do not understand how the puzzle pieces are grouped exactly.** \
> A1. Sorry for the confusion. Given that a video clip has eight frames, taking the temporal dimension as an example, we actually make an arbitrarily grouped permutation. For the permutation used in the main paper for concept explanation, i.e., {12345678} -> {84567123}, we make 3 groups of frames only as an example. However, since the permutations are not deterministic, different numbers of groups (e.g., 3, 4, or 5) will be selected for different video clips. More specifically, the number of groups is set according to our curriculum learning setting. The details are given in Appendix A.2.
>
> **Q2. Artificial patterns in the shuffled clips might be problematic.** \
> A2. Good question. Indeed it is a problem but it has been mitigated by our model design. First, we utilize color jittering to avoid learning shortcuts, which has been proved to have the ability of preventing the trivial learning. Second, we choose to solve the surrogate tasks instead of directly solving the puzzle, therefore being less affected by the artificial patterns. This is supported by Table 9. Third, in the experiments, we notice that pre-training with LCCD may mislead the model to learn from vertical and horizontal lines, but the other three tasks do not suffer from this trivial learning issue. When solving the four surrogate tasks jointly with adaptive weights (see Table 3), we can avoid learning the artificial patterns in our model. Please also see our response to AnonReviewer3.
>
> **Q3. Missing ablation experiments and analysis.** \
> A3. Thanks for pointing this out. (1) We choose 2 LCCs mainly for the designs of CLSC and CSPC. If we use one, three, or four, these two surrogate tasks would be too difficult to define. Note that non-local operation is just one way to fusing two features. We have also tried concatenation and adding, but the performance difference is smaller than 0.5%. (2) To learn with LCCD, the model needs to generate a mask and thus FPN is considered. If we drop FPN, we need to change the network structure significantly to generate masks. However, by looking at the ablation study results, we believe that dropping FPN would not affect too much because LCCD contributes the least among the four tasks. (3) We make a detailed ablation study on why and how each surrogate task performs in the appendix. The results show that each of them is indeed effective for solving CSJ. We also present the visualization results in Figure 4 to show how LCCD performs in pre-training.
>
> **Q4. For me, Table 3 is a bit hard to parse.** \
> A4. Thanks. We have done it in the revision.

---

### Official Review · AnonReviewer4 · 2020-10-21
**Performance is nice but lack insights**

**Rating:** 5
**Confidence:** 4

**Review:**

The paper presents a novel pretext task for self-supervised video representation learning (SSVRL). The authors design several surrogate tasks for tackling intentionally constructed constrained spatiotemporal jigsaw puzzles.  The learned representations during training to solve the surrogate tasks can be transferred to other video tasks. The proposed method shows superior performances than state-of-the-art SSVRL approaches on action recognition and video retrieval benchmarks.

## Strengths:

+. Good performances on two benchmarks.

+. Carefully designed surrogate tasks.

## Weaknesses:

-. Lack insightful analysis of how the idea is inspired, why it works. It seems the intuition of the paper is to make the 3D jigsaw problem easier to solve and it will just work. But why the easier problem could help learn better representations? Each of the two steps making the problem easier need to be analyzed more thoroughly: first, making the unconstrained jigsaw problem constrained; second, solving the surrogate tasks instead of solving the constrained jigsaw problem. Actually, the carefully designed surrogate tasks are quite different from the constrained jigsaw problem. They seems more ad-hoc but not a principled way to tackle the jigsaw problem. All these questions need more indepth clarification.

-. Experimental analysis is not thorough. In case the proposed method is not a principled method, but a carefully designed method. Extensive experiments of different variations of the proposed method could help better understand why the method works. A good performance on well-established benchmarks might be impressive, but analysis of why the performance can be achieved is more important.

-. Writing needs improvements. In the exposition of the proposed method section, some sentences are casual and misleading. For example, the third paragraph of sec 3.2. Besides, section 3.3 is a little bit difficult to follow. It could be possibly revised more concisely.

## Summary

Overall, the paper presents yet another method to design the pretext task for SSVRL. But my major concern is it lacks enough insights for inspiring future research for this topic. It might not be good enough for ICLR.

---

> ### Author Response · Authors · 2020-11-13
> **Response to AnonReviewer4**
>
> We’d like to thank the reviewer for the constructive comments and suggestions. We have accordingly improved our paper in the revision.
>
> **Q1. Lack insightful analysis of how the idea is inspired, why it works.** \
> A1. Thanks for pointing this out. Our work has been inspired by the intractability challenge in solving the unconstrained jigsaw problem. In (Kim et al. 2019), the authors claimed that "We generate a spatiotemporal cuboid consisting of 2x2x4 = 16 grid cells for each video. However, there are 16! possible permutations and include very similar permutations, which make the puzzle task very ambiguous. To avoid such ambiguity, we sample 4 crops instead of 16, in either spatial or temporal. More specifically, the crops are extracted from a 4-cell grid of shape 2x2x1 or 1x1x4." This means that directly solving a 3D puzzle turns out to be intractable. To overcome this challenge, we thus impose the proposed constraints on the 3D puzzle and introduce four surrogate tasks to make it more tractable. Our ablation study and visualization results (in the tables and figures) show that our proposed model indeed can learn better representation for video content understanding. Here, we’d like to emphasize the following two points: (1) As shown in Table 9, our choice of solving the surrogate tasks (instead of directly solving constrained jigsaw) can help to mitigate the trivial learning issue. (2) As illustrated in Figure 5, our CSJ is more effective in the temporal action segmentation task than the latest competitor MemDPC. That is, through solving the ‘true’ spatiotemporal jigsaw, our CSJ results in better understanding of the content of a long video. Additionally, more interesting works (e.g., with other surrogate tasks proposed) can be inspired by our CSJ model. We believe that we have made sufficient contributions to the SSVRL area.
>
> **Q2. Extensive experiments of different variations of the proposed method could help better understand why the method works.** \
> A2. Good suggestion! In addition to the ablation study in Table 3 of the main paper, we have also made a detailed ablation study for each surrogate task in the appendix. The reported results show that each surrogate task clearly contributes to solving the CSJ and our design/setting for each task is sound. The ablation study results of each surrogate task also show why it leads to performance improvements. For example, for CSPC, the spatial-only information, temporal-only information, and spatiotemporal information have all been explored in Table 7. The results show that exploiting more information for SSVRL consistently boosts the performance of our CSJ method. Moreover, the ablation study results in Table 9 show that solving the surrogate tasks is more effective than directly solving CSJ, indicating that the trivial learning issue in CSJ can be mitigated by solving the surrogate tasks jointly.
>
> **Q3. Writing needs improvements.** \
> A3. Thanks. We have made changes in the revision.

---

### Official Review · AnonReviewer3 · 2020-10-27
**Interesting SSL pretext task, unsure about how much it pushes the boundary of SSL**

**Rating:** 6
**Confidence:** 4

**Review:**

**Summary**
This paper proposes a new way of formulating and solving "spatiotemporal jigsaw puzzles", as a self-supervised pretext task for learning useful video representations. Positive results on two downstream tasks, action recognition and video retrieval, are shown.
The main contribution of the paper is a novel way of constraining the space of possible spatiotemporal permutations, in order to increase the tractability of the problem, and proposal of surrogate tasks that require learning and understanding of spatiotemporal continuity and correlations in order to solve them.

**Strengths**:
+ The pretext task of solving spatiotemporal jigsaw puzzles in order to learn meaningful video representations is well motivated, as has also been demonstrated in the literature.
+ Constraining the number of permutation to make the problem tractable indeed seems necessary, and the proposed method of doing so, along with the proposed surrogate tasks, are indeed effective, as indicated by the positive results on downstream tasks.
+ The ablation analysis performed shows a positive contribution of each of the described surrogate tasks and training scheme.

**Weaknesses**:

- I found some of the major statements in this work to be over-claimed:

  - " To our best knowledge, this is the first work on self-supervised video representation learning that leverages spatiotemporal jigsaw understanding". As you mention in Sec. 2,  Ahsan et al. (2019); Kim et al. (2019) both attempted to solve spatiotemporal jigsaw puzzles as a self-supervised pretexts task for learning spatiotemporal representation. This paper claims that the particular constraints imposed on the permutations used in those papers in order to increase the tractability of the problem, are not "true" spatiotemporal permutations. I believe this work at most relaxes some of those constraints, albeit in creative ways, but is not solving a fundamentally different problem.
 - Table 1, which demonstrates "state-of-the-art performance" on action recognition, is incomplete. Some stronger, not-included results of methods you did include in the table, and which, to the best of my knowledge, use only the RGB modality:
 Pace           |  S3D-G  |  87.1  |  52.6
 SpeedNet  |  S3D-G  |  81.1  |  48.8
for UCF101 (left) and HMDB51 (right).
  - For Table 2, which shows "new state-of-the-art in video retrieval", additional, stronger, "Pace" results exist :
Pace | C3D | 31.9 | 49.7 | 59.2 | 68.9 | 80.2
Pace | R(2+1)D | 25.6 |  42.7 |  51.3 |  61.3 |  74.0
for (L-R) top 1, 5, 10, 20, 50.

- For the visualization (Sec. 4.4), it would be nice to see what the network attends to in order to solve the pretext task, before fine-tuning on UCF101 to solve action recognition.


Out of curiosity -- often in self-supervised learning the network tends to learn "artificial cues" (such as boundary or compression artifacts) which help it solve the pretext task, without really learning anything meaningful. Significant work is usually required to mitigate such trivial learning. Did you have a similar problem? I can't seem to find any documentation of such a phenomenon in your work.

In general, I find new SSL work especially interesting if it (A) enables solving new tasks that were unfeasible before, and/or (B) pushes the boundary of SSL results on interesting/important tasks. Since (A) is, according to my understanding outlined above, not accurately demonstrated, and (B) is not shown, I vote for rejecting this paper in its current form. I would gladly reconsider given stronger results or the demonstration of newly enabled tasks based on the proposed method.


**Post-rebuttal**

I'd like to thank the authors for addressing my comments. I've read through the other reviews and responses, as well as the revised paper. The presented method for learning "true spatiotemporal permutations" is novel, and does indeed seem to learn effective representations.

What I'm not entirely sure about is how much this method manages to push the boundary of SSL. Comparing methods with different backbones is indeed tricky, and my intention was definitely not to discourage SSL works from academia. But the burden of proof should be on the new method to perform as close to an apples-to-apples comparison (in terms of backbone) to existing methods as possible. In the end, there are many many potential pretext tasks for SSL of video representations, and I do feel that in order to be publishable at a top-tier venue, they should either enable new tasks, or show clear superiority over existing methods.

Regarding temporal action segmentation as a newly enabled task -- I honestly missed this section, since it's in the appendix. This should be moved to the main paper.

If I could, I would be borderline on this paper. But since I can't, I'll give the authors the benefit of the doubt, and raise my rating to 6 (marginally above).

---

> ### Author Response · Authors · 2020-11-13
> **Response to AnonReviewer3**
>
> We’d like to thank the reviewer’s constructive comments. Our responses are detailed below.
>
> **Q1. Some of the major statements in this work are over-claimed.** \
> A1. Although there are several recent works on self-supervised video representation learning (SSVRL) using ‘spatiotemporal’ permutations, we are the first to propose ‘true’ spatiotemporal jigsaw for SSVRL. Specifically, in Kim et al. (2019), the authors claimed that "However, there are 16! possible permutations and include very similar permutations, which make the puzzle task very ambiguous. To avoid such ambiguity, we sample 4 crops instead of 16, in either spatial or temporal. More specifically, the crops are extracted from a 4-cell grid of shape 2x2x1 or 1x1x4." This means that the spatiotemporal jigsaw puzzle was actually solved with spatial-only and temporal-only permutations in Kim et al. (2019), but not with ‘true’ spatiotemporal permutations. Similarly, Ahsan et al. (2019) also didn’t exploited ‘true’ spatiotemporal permutations for SSVRL. In contrast, we provide a tractable but effective solution to spatiotemporal jigsaw understanding in this paper. Concretely, if we follow the statement in Kim et al. (2019), 16 crops are actually sampled from a grid of shape 2x2x4, i.e., 3D permutations are considered in our CSJ model. To our best knowledge, this is the first work on SSVRL that leverages ‘true’ spatiotemporal jigsaw understanding. Therefore we would like to stick the main claims but with more clarification on the context of the claims.
>
> **Q2. Tables 1, 2: incomplete comparison.** \
> A2. Thanks for pointing this out. Note that we have already included the two methods (i.e., Pace and SpeedNet) mentioned by the reviewer in Tables 1, 2. Concretely, in Table 1, Pace reports the result with R(2+1)D, and SpeedNet reports the result with I3D. Even if both R(2+1)D and I3D are slightly stronger than our R2D3D backbone, our model still achieves better overall performance than Pace and SpeedNet. It is worth pointing out that since S3D-G is much stronger than our backbone R2D3D, it is unfair to compare our CSJ method to Pace and SpeedNet using S3D-G. In Table 2, Pace reports the result with R3D, and SpeedNet reports the result with S3D-G. Note that R3D has a similar structure to R2D3D, and S3D-G is much stronger than our backbone. Our model also outperforms Pace and SpeedNet in the retrieval task. In our ongoing work, when more GPUs can be made accessible, we will employ S3D-G as our backbone.
>
> **Q3. It would be nice to see what the network attends to in order to solve the pretext task, before fine-tuning on UCF101 to solve action recognition.** \
> A3. Thanks. We have shown the visualization of the LCCD predictions from the pre-trained models in Figure 4. We can observe that solving the LCCD task indeed enables us to learn the locations of LCCs and thus understand the spatiotemporal continuity, which is a key step towards video content understanding.
>
> **Q4. Significant work is usually required to mitigate such trivial learning. Did you have a similar problem?** \
> A4. Good question. This trivial learning problem indeed exists but has been mitigated by our model design. More specifically: (1) Following MemDPC, we utilize color jittering to prevent the network from learning artificial cues. Color jittering has been proved to have the ability of preventing learning shortcuts. (2) Learning with CSJ would result in artificial patterns, as pointed out by the reviewer. Therefore, instead of directly solving CSJ, we choose to solve the surrogate tasks (see the support results in Table 9). (3) In the experiments, we noticed that pre-training only with LCCD may mislead our model to learn from vertical and horizontal lines, but the other three tasks do not suffer from this trivial learning issue (see Table 3). When solving the four surrogate tasks jointly with adaptive weights, we are able to avoid learning the artificial patterns in our model.
>
> **Q5. In general, I find new SSL work especially interesting if it (A) enables solving new tasks that were unfeasible before, and/or (B) pushes the boundary of SSL results on interesting/important tasks.** \
> A5. Thanks for pointing this out. But we disagree with this comment. For the point (A), we have applied our CSJ to the temporal action segmentation task. The qualitative results in Figure 5 clearly show that our CSJ is more effective in this new downstream task than the latest competitor MemDPC. That is, through solving the ‘true’ spatiotemporal jigsaw, our CSJ results in better understanding of the content of a long video. For the point (B), only with a weak backbone R2D3D (due to the limited number of GPUs at hand), our CSJ model still outperforms the latest competitors (e.g., Pace and SpeedNet) with similar backbones in the action recognition and retrieval tasks. Taking both points into account, we believe we have made sufficient contributions to the SSVRL area.

---

### Official Review · AnonReviewer1 · 2020-10-27
**A sound paper on 3D Jigsaw in Video**

**Rating:** 6
**Confidence:** 4

**Review:**

**Summary and contributions**:
The authors propose an approach for solving a constraint version of the 3D (space+time) jigsaw puzzle over a video, using four easier surrogate tasks. The four surrogate tasks are diverse, having different formulations: regression (LCCD, CCMR), classification (CSPC), and noise contrasting (CLSC) problems. All of them are learned together, using a joint learning objective.

The authors show the value of the newly learned representations in the self-supervised scenario on two downstream tasks (video action recognition and video retrieval), achieving state-of-the-art results compared with new methods.

**Strengths**:
- The paper comes with a solution for expanding a classical self-supervised 2D problem formulation to 3D, proposing a tractable approach by breaking it into four tasks and imposing constraints through them.
- A good amount of details on the method and on how the permutations and the surrogate tasks were chosen.
- The fact that even though the permutations are heavily constrained, they proved to be useful.
- The newly learned representations, that embed the temporal and spatial continuity aspects, achieve state-of-the-art results on two video tasks.

**Weaknesses**:
- The temporal aspect is not sufficiently highlighted in the experiments:
- An ablation study to better distinguish between the spatial and spatiotemporal representations, with both quantitative and qualitative experiments would strengthen the submission.
- It would be interesting to see how much the quantity of temporal information reflects in the performance (ablation on the number of used frames)

**Quality**:
The idea is simple but complex enough to generate valuable representations, having the temporal aspect integrated. The paper is technically sound.

**Clarity**:
The paper is clearly written and easy to read and follow.

**Novelty**:
The overall idea is not novel, but the way it is implemented and the proposed constraints are novel.

**Significance of this work**:
This work is relevant for the field, it incrementally advances the current integration of the temporal and spatial aspects in video.

**Typos**:
The 3rd vertical line in Table 3 should be shifted with 1 column.

---

> ### Author Response · Authors · 2020-11-13
> **Response to AnonReviewer1**
>
> We’d like to thank the reviewer for the constructive comments and suggestions. We have accordingly made the changes in the revision.
>
> **Q1. The temporal aspect is not sufficiently highlighted in the experiments.** \
> A1. Thanks for pointing this out. Our explanations are four-fold: (1) According to Table 1, our CSJ method, using both spatial and temporal information, achieves a significant improvement of 15.1% over the latest spatial-only method CMC (Tian et al. 2020) on UCF-101. This clearly shows the effectiveness of the temporal part of our CSJ method for self-supervised video representation learning (SSVRL). (2) The spatial-only information, temporal-only information, and spatiotemporal information have all been explored in the CSPC task. According to Table 7, exploiting more information for SSVRL generally leads to better results, again showing the effectiveness of the temporal part of our CSJ method. (3) Note that each frame’s attention map in Figure 3 actually involves four adjacent frames. Therefore, we can observe from Figure 3 that both spatial and temporal information is captured by our CSJ method (see the ‘Playing Flute’ example). This provides further evidence that our CSJ method indeed can learn better spatiotemporal representation. (4) We have also applied our CSJ method to the temporal action segmentation task. The qualitative results in Figure 5 clearly show that our CSJ is more effective in this downstream task than the latest competitor MemDPC. In summary, through solving the ‘true’ spatiotemporal jigsaw, our CSJ results in better understanding of both spatial and temporal content of a long video.
>
> **Q2. An ablation study to better distinguish between the spatial and spatiotemporal representations, with both quantitative and qualitative experiments would strengthen the submission.** \
> A2. Thanks for the suggestion. First, the spatial-only information, temporal-only information, and spatiotemporal information have all been explored in the CSPC task in Table 7. The quantitative results show that exploiting more information for SSVRL consistently boosts the performance of our CSJ method. Second, each frame’s attention map actually involves four adjacent frames in Figure 3. Therefore, the qualitative results show that both spatial and temporal information can be well captured by our CSJ method (see the ‘Playing Flute’ example). Please also see our response to Q1.
>
> **Q3. It would be interesting to see how much the quantity of temporal information reflects in the performance (ablation on the number of used frames).** \
> A3. Good suggestion! The 16-frame setting has been widely used by existing SSVRL methods. In this paper, for fair comparison, only 16 frames are used in our CSJ. We will explore more frames in our CSJ, when more GPUs become accessible to us. Moreover, the qualitative results in Figure 5 show that our CSJ is also effective for coping with long videos (with many frames) in the temporal action segmentation task.
>
> **Q4. The 3rd vertical line in Table 3 should be shifted with 1 column.** \
> A4. Thanks. We have done it in the revision.

---

### Author Response · Authors · 2020-11-14
**Major changes and contribution claim for Paper508**

Thanks all reviewers for their constructive comments. We have made the following changes in the revision: (1) Figure 4 is added to show the visualization of the LCCD prediction. (2) Figure 5 is added to show the qualitative results of temporal action segmentation. (3) Table 9 is added to show the comparative results between directly solving CSJ and solving the surrogate tasks instead. (4) Table 3 is reorganized for easy understanding. (5) Some claims are reworded and our paper is carefully polished.

We are glad to see that most of the reviewers found some strengths of our paper. To help the reviewers to make a decision, we summarize our main contributions as follows: (1) We introduced a new pretext task called Constrained Spatiotemporal Jigsaw (CSJ) for self-supervised video representation learning (SSVRL). To our best knowledge, this is the first work on SSVRL that leverages ‘true’ spatiotemporal jigsaw understanding. (2) We proposed a novel constrained shuffling method to construct easy 3D jigsaws that contain large LCCs. Four surrogate tasks are then formulated in place of the original jigsaw solving tasks. They are much more solvable yet remain effective in learning spatiotemporal discriminative representations. Importantly, solving them jointly can help to mitigate the trivial learning issue in SSVRL. (3) Extensive quantitative and qualitative results have shown why our CSJ model is effective for SSVRL.

Since only a weak backbone R2D3D is used in our CSJ model due to the limited GPU resources available to us at academia, we hope that the reviewers could pay more attention to the contribution/novelty of our work (rather than comparing the SOTA results obtained with much larger backbone, more training data and more modalities) when making a decision. We believe that this can inspire more researchers in the field to develop spatiotemporal pretext tasks for self-supervised learning in videos.

---

### Decision · Program_Chairs · 2021-01-07
**Final Decision**

**Decision:**

Reject

**Comment:**

This paper presents a new idea and approach for self-supervised video representation learning.

The reviewers' opinions diverge. R1 suggests that the paper is (marginally) above the threshold. R2 supports the paper, saying that he/she likes the idea behind the paper. R3 explicitly mentioned that he/she would like to provide a borderline rating (but cannot due to the system). R4 is not in favor of the paper, even after the rebuttal.

The AC’s opinion is more aligned with R3 and R4, who are the more senior reviewers among the four. There are two main concerns with the paper: technical contribution and experimental comparison. In terms of the contribution, both the reviewers find that the paper is lacking: "What I'm not entirely sure about is how much this method manages to push the boundary of SSL." (by R3). "Overall, the paper presents yet another method to design the pretext task for SSVRL. But my major concern is it lacks enough insights for inspiring future research for this topic." (by R4). The authors argue the difficulties of being in academia in doing this research with limited computation resources and argue that the reviewers should focus more on novelty and contribution, but even after the rebuttal, R4 is not convinced and R3 is still mildly concerned whether the proposed approach really brings something new to the field as the paper fails to show "clear superiority over existing methods".

In addition, as pointed out by the reviewers, there are several state-of-the-art self-supervised video representation learning works that the paper misses to cite, or compare against. In addition to Pace and SpeedNet R3 mentioned, below are approaches reporting results on UCF101 and HMDB with the standard self-supervised classification task setting (Table 1):

AVTS 89.0, 61.6
CVRL 92.1, 65.4
ELo 93.8, 67.4
XDC 94.2, 67.4
GDT 95.2, 72.8

We note that all these results are much superior to the best results reported by the proposed approach, 79.5 and 50.9 on UCF101 and HMDB. The authors mention in the rebuttal that these superior approaches not included in the paper use stronger backbones (and are thus omitted), but we believe a more academically proper attitude is to include all these numbers and explicitly describe why the proposed approach is not performing better, instead of completely omitting their results.

The AC also questions whether the R2D3D-34 backbone used in this paper really is computationally lighter compared to the backbones used in previous approaches like R(2+1)D-18, which alternates 2D residual modules and 2+1D residual modules (using much fewer parameters and compute than 3D modules) and also has fewer layers. XDC using R(2+1)D-18 backbone reports 86.8/52.6 (UCF/HMDB) accuracies with Kinetics-400 unlabeled data. AVTS also reports 84.1/52.5 (UCF/HMDB) accuracies using MC3-18 backbone. Similarly, GDT uses R(2+1)D-18 backbone and reports 89.3/60.0 (UCF/HMDB) accuracies using unlabeled Kinetics-400.  Even MemDPC reports 86.1/54.5 using R-2D3D backbone when optical flow feature is added. All these are far superior to the results being reported in the paper.

Overall, we view the experimental section of this paper as incomplete, and we cannot convince ourselves that the paper reaches the quality of ICLR.


[AVTS] Korbar, B., Tran, D., Torresani, L.: Cooperative learning of audio and video models
from self-supervised synchronization. In: NeurIPS (2018)

[ELo] A. Piergiovanni, A. Angelova, and M. S. Ryoo. Evolving losses for unsupervised video representation learning. In Proc. CVPR, 2020

[XDC] H. Alwassel, D. Mahajan, L. Torresani, B. Ghanem, and D. Tran. Self-supervised learning by cross-modal audio-video clustering. arXiv preprint arXiv:1911.12667, 2019

[GDT] M. Patrick, Y. M. Asano, R. Fong, J. F. Henriques, G. Zweig, and A. Vedaldi. Multi-modal self-supervision from generalized data transformations. arXiv preprint arXiv:2003.04298, 2020

[CVRL] R. Qian, T. Meng, B. Gong, M.-H. Yang, H. Wang, S. Belongie, and Y. Cui. Spatiotemporal contrastive video representation learning. arXiv preprint arXiv:2008.03800, 2020